# Deep Learning-Based Segmentation of the Ulnar Nerve in Ultrasound Images

**DOI:** 10.3390/medicina62010113

**Published:** 2026-01-05

**Authors:** Matthew Bailey Webster, Ko Eun Kim, Yong Jae Na, Joonnyong Lee, Beom Suk Kim

**Affiliations:** 1Chung-Ang University Industry-Academic Cooperation Foundation, Seoul 06974, Republic of Korea; 2Mellowing Factory Co., Ltd., Seoul 03726, Republic of Korea; 3Department of Dermatology, Korea University Guro Hospital, Seoul 08308, Republic of Korea; 4Department of Physical and Rehabilitation Medicine, Chung-Ang University Gwangmyeong Hospital, Gwangmyeong-si 14241, Republic of Korea; 5Department of Physical and Rehabilitation Medicine, Chung-Ang University College of Medicine, Seoul 06974, Republic of Korea

**Keywords:** ultrasonography, peripheral nerves, ulnar nerve, deep learning

## Abstract

*Background and Objectives*: We evaluate deep learning-based segmentation methods for detecting the ulnar nerve in ultrasound (US) images, leveraging the first-ever large US dataset of the ulnar nerve. We compare several widely used segmentation models, analyze their performance, and evaluate several common data augmentation techniques for the US. *Materials and Methods*: Our analysis is conducted on a large dataset of 4789 US images from 545 patients, with expert-annotated ground-truth segmentations of the ulnar nerve, and uses six segmentation models with several backbone architectures. Further, we analyze the statistical significance of five common data augmentation techniques on segmentation performance: flipping, rotation, shearing, contrast and brightness adjustments, and resizing. *Results*: In this study, the shear, rotate, and resize augmentations consistently improved segmentation performance across multiple runs, with *p*-values < 0.05 in a paired *t*-test relative to the no-augmentation baseline. Furthermore, we showed that newer architectures do not provide any metric improvements over traditional U-Net models, which achieved a Dice score of 0.88 and an IoU of 0.81. *Conclusions*: Through our systematic analysis of segmentation models and data augmentation strategies, we provide key insights into optimizing deep learning approaches for ulnar nerve segmentation and other US-based nerve segmentation tasks.

## 1. Introduction

High-resolution ultrasound (US) is vital in neuromuscular medicine for diagnosis, treatment, and prognostication. It identifies peripheral nerve structures, measures nerve parameters, and guides real-time interventions. The US is beneficial for diagnosing entrapment neuropathies, with guidelines indicating that the median nerve’s cross-sectional area (CSA) measurement at the wrist provides level A evidence for carpal tunnel syndrome (CTS) [1,2,3,4]. This has led to rapid expansion in its clinical use. Furthermore, the ulnar nerve, particularly at the elbow, is the second most common site of upper extremity entrapment neuropathy, following the median nerve at the wrist [2]. However, the efficacy of deep learning approaches for ulnar nerve segmentation in US remains understudied within the literature.

Peripheral nerves exhibit characteristic US features: appearing as linear hypoechoic areas separated by hyperechoic bands in longitudinal scans and as a honeycomb-like structure in axial scans [5,6,7]. However, accurate identification can be difficult due to anatomical variability, individual echotexture differences, and similarity to tendons, potentially leading to diagnostic errors and malpractice. Clinicians differentiate nerves using strategies such as assessing echotexture stability despite probe tilting (to account for anisotropy) and by tracing nerve pathways [6,8]. Still, these methods are time-consuming, labor-intensive, and require extensive training. Manual measurements further stress the need for automated solutions to boost efficiency and accuracy.

Deep learning-based segmentation models have become a crucial part of many image processing workflows and enhance computer-aided diagnostics (CAD) [9,10,11,12,13]. These models have proven useful in several US imaging applications ranging from breast cancer [14,15] and vessel segmentation [16,17] to peripheral nerve segmentation [18,19]. However, US segmentation remains challenging due to the often small size of target structures and the presence of noise and distortion artifacts. Obtaining high-quality images with expert-annotated segmentation labels is labor-intensive and time-consuming, usually limiting prior studies to small datasets. These dataset constraints introduce distribution bias, reducing the real-world generalizability.

Despite advances in deep learning for medical imaging, ultrasound-based peripheral nerve segmentation remains challenging. Small datasets have limited model generalizability and require restricted parameter sizes to prevent overfitting. Data augmentation for US imaging has also received insufficient attention. Techniques like shearing and resizing may improve model robustness, but their effects are not well understood. Ulnar nerve segmentation, in particular, is underrepresented in the literature, with few comprehensive evaluations of segmentation models or augmentation strategies. To address these gaps, our study compares and benchmarks deep learning segmentation models for ulnar nerve detection, using the largest expert-annotated dataset to date, comprising 4789 ultrasound images from 545 patients. We also assess the impact of various augmentation methods, including rotation, shearing, resizing, cropping, and adjustments to contrast and brightness. This work provides practical guidance for applying deep learning to peripheral nerve segmentation, with a focus on the understudied ulnar nerve.

## 2. Materials and Methods

We compare six common segmentation architectures, each using three different backbone architectures as feature encoders. Further, we explore the impact of common data augmentation methods applied to US images on the segmentation of the ulnar nerve.

### 2.1. Deep Learning Architectures

The six segmentation network architectures studied for ulnar nerve segmentation are U-Net [11], LinkNet [20], U-Net++ [21], DeepLabV3+ [22], Pyramid Attention Network [13], and Segformer [12]. U-Net++ improves upon U-Net by embedding nested, dense skip pathways, thereby reducing the semantic gap between the encoder and decoder layers. While this enhances multi-scale context capture, it significantly increases complexity and parameter count. LinkNet, in contrast, directly connects encoder inputs to corresponding decoder outputs, preserving high-resolution spatial details during upsampling while minimizing computational overhead. The Pyramid Attention Network (PAN) introduces a multi-scale attention module that aggregates features across various scales to emphasize informative regions and suppress irrelevant ones. Computational efficiency is crucial in ultrasound segmentation, as real-time assistance is essential for clinical applications [23,24]. DeepLabV3+ employs an encoder–decoder structure with atrous separable convolutions in both the Atrous Spatial Pyramid Pooling (ASPP) module and decoder, refining segmentation boundaries while maintaining computational efficiency. Segformer further optimizes the balance between performance and efficiency by leveraging transformer-based architectures, decoupling encoding and decoding to achieve robust global context modeling with minimal computational cost.

In addition, we employ various encoder architectures, each with distinct computational trade-offs, including ResNet [25], EfficientNet [26], and Mix Transformer (MiT) encoders from the Segformer framework. Table 1 summarizes the computational cost in MACs (multiply–accumulate) for a 512 × 512 image and the parameter counts across the different combinations of backbones and architectures considered in this study. Although our dataset is relatively large compared to prior studies, it remains small compared to natural image datasets. To mitigate overfitting and reduce compute costs, we use the more minor backbone variants.

### 2.2. Data Augmentations

Various data augmentation schemes have been proposed for use with US images including random resizing, cropping, rotation, shearing and more. We selected five data augmentation techniques for analysis: random rotation, resizing, contrast and brightness adjustments, flipping, and shearing. Random cropping was always applied during training to standardize image sizes for batching and encourage learning from different regions, enhancing focus on key anatomical features. Random resizing simulates variability in object size and probe distance, improving generalization across imaging conditions. Contrast and brightness adjustments mimic natural fluctuations in ultrasound quality due to machine settings and patient factors. Flipping, rotation, and shearing introduce orientation and distortion variations, reflecting probe movement and operator variability. Together, these augmentations enhance data diversity, reduce overfitting, and improve model generalizability for ultrasound image analysis.

### 2.3. Dataset Description

A computerized search of electronic medical records and picture archiving and communication systems was performed to identify patients who underwent US examinations of the upper extremities from March 2019 to December 2024. Among the 983 initially identified subjects, 545 subjects over 20 years of age with normal findings in ulnar nerve conduction studies were included. From the 545 patients, a total of 4789 US images are included as a part of this study. Eighty percent of the patients are randomly selected for training and validation (a total of 436 patients) and the remaining patients are withheld for the test set (a total of 109 patients).

An expert clinician with over 15 years of clinical experience in neuromuscular ultrasound conducted all examinations and annotations using the miniSONO (Alpinion Medical Systems, Seoul, Republic of Korea). Subjects were placed in the supine position with elbows flexed to 90 degrees, forearms fully supinated, and fingers maintained in slight flexion. To accurately identify the ulnar nerve, the clinician performed dynamic ultrasound scanning by slowly moving the transducer proximally and distally along its course. Ultrasound images of the ulnar nerve were acquired at five standardized locations per extremity: specifically, five images were captured at 1 cm intervals, spanning from 2 cm proximal to 2 cm distal to the medial epicondyle at the elbow. The examiner endeavored to capture the clearest and most representative images of the nerve by orienting the ultrasound beam perpendicular to the course of the nerve. Nerve boundaries were defined using the inner epineurial contour, excluding the outer hyperechoic rim. Ambiguous regions—where the perineurium blended with adjacent hyperechoic tissue—were resolved by reviewing adjacent frames to maintain consistent labeling.

Images with significant motion artifacts, acoustic shadowing, signal dropout, or excessive deformation caused by probe pressure were excluded from the dataset. All included images underwent expert quality review to ensure sufficient visualization of the ulnar nerve before annotation.

### 2.4. Outcome Variables

In this study, the performance of the deep learning architectures was verified using the following indicators: recall, precision, F1 score, Dice score, and intersection over union (IoU).(1)$$O(Dice)=\frac{2\cdot \left|X\cap Y\right|}{\left|X\right|+\left|Y\right|}=\frac{2\cdot TP}{2\cdot TP+FP+FN}$$(2)$$(IoU)=\frac{\left|X\cap Y\right|}{\left|X\cup Y\right|}=\frac{TP}{TP+FP+FN}$$
where *X* and *Y* represent the predicted and ground-truth segmentation masks, respectively. Further, TP is the true positive rate, FP is the false positive rate, and FN is the false negative rate.

### 2.5. Implementation Details

For the experiments in this study, we used the PyTorch (2.5.1) machine learning framework, along with the segmentation-models-pytorch library, to ensure reproducible implementations of each segmentation model [27]. We adopt the Dice loss function, denoted by L, to find the boundaries of peripheral nerves in the US images. The Dice loss function *L* is defined as follows:*L* = 1 − (Dice)(3)

(Dice) is defined in Equation (1). The Dice loss can alleviate blurred boundary problems often observed with cross-entropy loss and is also effective at handling class imbalance, i.e., the nerves occupy only a small portion of the US image.

All models were trained using the Adam optimizer with a learning rate of 1 × 10^−4^ and a weight decay of 1 × 10^−6^. Training was conducted for 150 epochs, with the learning rate reduced to 1 × 10^−5^ after 100 epochs. During training, images were either padded or randomly cropped to 512 × 512 before applying data augmentations, with resizing applied first. A batch size of 32 was used for all experiments, and all models were trained on an NVIDIA RTX A6000 GPU.

To avoid data leakage, all dataset splits were performed on a patient-wise basis, such that images from the same subject were not shared across the training, validation, and test sets. Data augmentations were applied exclusively during training, while the validation set was used to monitor model convergence and guide learning-rate scheduling.

Data augmentations were applied in the following order: (1) random resize and cropping, (2) random contrast and brightness adjustments, (3) random rotations, (4) random shearing, and (5) random horizontal flipping. Resizes were chosen uniformly between a 25% decrease and a 25% increase in the image’s height, preserving the aspect ratio. Brightness and contrast adjustments were selected uniformly within ±50% of their original values. Rotations and shearing transformations were applied uniformly between −15° and 15°, while horizontal flips were performed with a 50% probability.

## 3. Results

### 3.1. Quantitative

To evaluate the impact of different data augmentation strategies, we trained a U-Net model with a ResNet-34 backbone using every possible combination of augmentations. Figure 1 presents the test-set Dice scores for all single augmentation strategies, along with no augmentation and all augmentations. Our results indicate that all augmentations, combined, provided a statistically significant improvement in the Dice score on our test set. Further, in Figure 2, we see that as the number of data augmentations increases, the Dice score also increases. For our augmentation analysis, we employ Bonferroni correction to our paired *t*-tests. Bonferroni correction is a statistical adjustment that accounts for the increased risk of false positives when performing multiple comparisons simultaneously. It works by dividing the significance threshold by the number of tests being conducted. For both Figure 1 and Figure 2 the overall pattern of statistical significance did not change after applying this correction. The results in Figure 1 show the impact of individual augmentations, with the most significant being the rotation augmentation, while Figure 2 shows that the number of combinations employed is also statistically significant in improving overall segmentation performance.

Table 2 shows that despite extensive testing, the U-Net with a ResNet43 backbone achieved the highest Dice score and IoU. Multi-scale models like PAN and DeepLabV3+ likely provided no advantage because random resizing already enforced scale invariance, and the ulnar nerve’s boundaries are not complex enough to benefit from such processing. Additionally, newer state-of-the-art models like Segformer did not show a noticeable improvement over U-Net, suggesting that appropriate model capacity and effective data augmentation strategies are sufficient for achieving state-of-the-art results in ulnar nerve segmentation. However, DeepLabV3+ with EfficientNet backbones performed comparably to U-Net at a lower computational cost, making them a viable alternative for compute-constrained applications despite a slight performance trade-off.

### 3.2. Qualitative

For our qualitative analysis, we use predictions from the best-performing model identified in our architecture search: a U-Net with a ResNet34 backbone. Figure 3 presents general segmentation results for ulnar nerve images captured by the miniSONO device. Ground-truth regions are outlined in green, while predicted areas are shown in red. Failure cases fall into three main categories. First, as shown in Figure 4a, some predictions only partially overlap with the ground truth. This may occur when the model misinterprets an internal nerve structure, possibly mistaking part of the perineurium (the sheath surrounding a nerve fascicle) for the outer nerve boundary. As a result, the predicted segmentation may erroneously pass through the middle of the nerve.

Second, as shown in Figure 4b, some segmentations deviate from the expected elliptical shape. In clinical practice, the ulnar nerve typically appears as an ellipse, and uneven margins should still approximate this form. However, in some cases, the model over-segments, including adjacent structures such as small vessels.

Third, Figure 4c illustrates a rare but significant failure: a total miss where the predicted segmentation does not overlap with the ground truth. In such cases, both precision and recall drop to zero. Although infrequent, only accounting for 0.6% of all results on the test set, these failures highlight potential limitations that could impact clinical applications.

## 4. Discussion

This study evaluated deep learning-based segmentation methods for ulnar nerve detection in US images, addressing key challenges in neuromuscular medicine. Using a robust dataset of 4789 US images from 545 patients, we compared various segmentation architectures and data augmentation techniques in Section 2.1 and Section 2.2, respectively. Through these experiments, we show that a sufficient dataset size, data augmentations, and traditional segmentation architectures, such as U-Net, provide reasonable performance in the segmentation of the ulnar nerve.

Since its development, U-Net [11] has been widely adopted for ultrasound (US) image segmentation in various applications, including the left ventricle [28], endometrium [29], breast [30], fetal skull [31], cerebral ventricles [32], and prostate [33,34]. U-Net has also been extensively applied to US nerve segmentation, particularly in regional anesthesia. Baby and Jereesh [35] used U-Net for brachial plexus nerve detection, while Kakade and Dumbali [36] combined Linear Gabor Binary Patterns for preprocessing with U-Net and Principal Component Analysis (PCA) for postprocessing. Other studies have introduced architectural modifications to enhance the U-Net’s performance. Wang et al. [37] introduced ResUNet, which incorporates residual modules, dense atrous convolutions, and multi-kernel pooling. Weng et al. [38] applied Neural Architecture Search (NAS) to optimize the U-Net’s backbone for US nerve segmentation. Horng et al. [19] developed DeepNerve, which integrates an LSTM into the U-Net’s bottom layer, using a MaskTrack scheme to improve median nerve tracking in sequential US video frames. Beyond these adaptations, various U-Net variants have been proposed for broader biomedical segmentation tasks. U-Net++ [21] incorporates dense blocks to enhance skip connections. Attention U-Net [39] introduces attention gates to focus on salient features while suppressing irrelevant regions. Scale-attentional U-Net [40] adds gating to cascaded convolutions without non-linearities or pooling at each block, allowing for dynamic adjustment of the receptive field throughout the network. Despite these advances, few studies have systematically compared segmentation models for ulnar nerve detection, particularly with respect to computational efficiency and data augmentation.

Rather than proposing a specialized architecture, this study systematically analyzed common data augmentations and segmentation networks on a large, expert-labeled US dataset. Data augmentation provided modest but consistent improvements. Random rotation augmentation yielded the most statistically significant performance improvement over the no-data-augmentation baseline (in a paired *t*-test), likely because it simulates common variations in probe orientation during ultrasound examinations. Random resizing and random shearing emerged as the second-most-effective augmentations, potentially aiding generalization across patients—a factor independent of operator technique or equipment. Contrast and brightness adjustments showed no significant effect in our study. Since all data were acquired using a single device, the miniSONO, the limited impact of this augmentation is unsurprising. Flipping augmentation had a larger effect than contrast and brightness augmentation. Still, the difference was not statistically significant, possibly due to factors that were consistent across operator techniques and device characteristics. As expected, augmentation effects were moderate given the large dataset, consistent with prior research showing diminishing returns with increased data [41]. Despite testing modern architectures, the original U-Net with a ResNet34 backbone achieved the highest performance (Dice: 0.8808, IoU: 0.8063), suggesting that traditional models remain competitive when well implemented. Shown in Figure 4 are some representative cases of poor predictions, including partial overlaps, misinterpretation of internal nerve structures, non-elliptical segmentations, and rare total misses. While our dataset was larger than most prior US studies, all images were captured on a single device (miniSONO). Future work should assess performance across multiple US machines for broader generalizability and explore shape priors or anatomical constraints to mitigate failure modes.

This study has significant merit in applying deep learning techniques to the ulnar nerve, one of the most frequently affected peripheral nerves in the upper extremity, for the treatment of neuropathy. We conducted the first comprehensive research using a large volume of high-quality peripheral nerve ultrasound data, demonstrating the successful application of deep learning to the ulnar nerve and achieving high accuracy in nerve identification while complementing previous work on the median nerve. The encouraging results suggest promising implications for developing sophisticated software capable of peripheral nerve identification. We expect that establishing a robust methodological framework will lay the critical groundwork for advanced neurological imaging technologies. Future research should expand our approach to additional clinically significant nerves in both the upper and lower extremities and systematically develop algorithms to distinguish between normal and abnormal neural structures, which will be essential for translating these findings into clinical diagnostic tools.

Most previous deep learning studies on peripheral nerve ultrasound segmentation have focused on the median nerve, with reported Dice coefficients generally ranging from 0.80 to 0.90, depending on the model architecture and dataset characteristics [18,19,35,36,40]. These studies, which include approaches such as U-Net, ResUNet, NAS-optimized U-Net, and DeepNerve, collectively highlight that the median nerve has been the predominant focus of segmentation research. In our study, the best-performing model achieved a Dice score of 0.8808 and an IoU of 0.8063, placing its performance within the upper range of these prior median nerve studies. Notably, despite the ulnar nerve exhibiting greater anatomical variability and a more heterogeneous echotexture than the median nerve, our model achieved a comparable level of accuracy. This suggests that, given sufficient dataset size and effective augmentation strategies, ulnar nerve segmentation can achieve performance on par with that reported for the median nerve. Such contextual comparison underscores the contribution of our work in addressing the relative scarcity of ulnar nerve segmentation research. It supports the potential development of more generalized multi-nerve segmentation frameworks in future studies.

The accuracy achieved by our model (Dice 0.8808) is comparable to, and in the upper range of, the inter-observer variability reported in neuromuscular ultrasound studies. Prior work has shown that manual delineation of nerve boundaries by different clinicians typically yields Dice scores of approximately 0.80–0.88, indicating that a degree of variability is inherent to human interpretation. In this context, the segmentation performance of our model reaches the level generally considered sufficient to support clinically meaningful tasks, such as automated CSA estimation or detection of focal nerve enlargement. Nevertheless, segmentation accuracy alone does not ensure diagnostic reliability, and future studies should evaluate the clinical performance of CSA measurements derived from automated segmentation.

Despite the promising performance of our segmentation model, several intrinsic characteristics of peripheral nerve US must be acknowledged when interpreting the results. In clinical practice, distinguishing the precise number and arrangement of fascicles on ultrasound is often tricky, as the echotexture of peripheral nerves varies considerably across individuals and anatomical regions. The boundaries between the epineurium and perineurium are frequently indistinct, and the transition from neural tissue to surrounding connective tissue is often gradual rather than sharply demarcated. These ambiguities can make it difficult to delineate the nerve’s outer contour accurately. As illustrated in the first example of Figure 4, the model occasionally misinterprets portions of the perineurium or adjacent hyperechoic structures as part of the nerve margin, resulting in segmentation that partially bisects the nerve or deviates from its expected elliptical morphology. Such cases highlight the fundamental limitations of ultrasound imaging in resolving delicate intraneural structures.

Recent multimodal studies integrating histology with high-resolution ultrasound have shown that many structures appearing as single fascicles on ultrasound actually correspond to histologically identified clusters of multiple fascicles. That epineurial or perineurial boundaries may shift or blend depending on scanning angle, probe pressure, and tissue composition [42]. These findings suggest that leveraging histological ground truth improves the interpretability and robustness of deep-learning models applied to nerve imaging. Incorporating multimodal supervision—such as histologic cross-sections, MR microscopy–derived fascicular maps, or other microstructural references—could enable future models to distinguish actual fascicular boundaries better, avoid erroneous perineurial segmentation, and more closely approximate the microanatomy of peripheral nerves.

Accordingly, future work should prioritize multimodal fusion approaches that bridge histological insights with ultrasound-based AI segmentation. Such studies have the potential not only to overcome the intrinsic resolution limits of ultrasound but also to establish biologically meaningful constraints that guide model predictions. Ultimately, integrating histologic validation into deep-learning development may yield segmentation systems capable of supporting fascicle-level analysis and more accurate assessment of neuropathic changes, thereby expanding the clinical relevance of automated peripheral nerve imaging.

Although our model demonstrated high segmentation accuracy, several considerations remain before it can be deployed in real-world clinical environments. Ultrasound imaging varies substantially across devices, transducer types, and operator techniques, which may affect model performance outside of the controlled setting of this study, where all images were acquired using a single device and a single examiner. Additionally, real-time inference capability and integration into existing clinical workflows must be evaluated for practical use. Further validation in diverse patient populations, particularly in those with neuropathies that alter nerve morphology, is essential to determine whether the model can reliably support clinical decision-making. Establishing such evidence will be a critical step toward translating automated ulnar nerve segmentation into routine neuromuscular ultrasound practice.

Despite the strengths of this study, several limitations must be acknowledged. First, the dataset was acquired exclusively using a single ultrasound device (miniSONO) and by a single expert examiner. Although this consistency reduces variability and improves annotation quality, it may limit the model’s generalizability to other devices or operators. Because a single examiner generated all annotations, the ground truth does not capture inter-observer variability, which is inherent to neuromuscular ultrasound interpretation. Future studies should incorporate multiple annotators or consensus labeling to establish more robust and generalizable reference standards. Second, the dataset included only patients with normal ulnar nerve conduction studies. Hence, the model’s ability to detect or segment pathological nerves, such as in cases of ulnar neuropathy or entrapment syndromes, remains untested. Finally, the study did not assess clinical endpoints, such as the accuracy of cross-sectional area (CSA) measurements or diagnostic utility, which are critical for real-world deployment in neuromuscular practice. Although Dice performance was high, segmentation accuracy does not necessarily translate to accurate CSA estimation, and future validation studies are required to assess the clinical reliability of automated CSA measurements.

## 5. Conclusions

Our comprehensive evaluation of deep learning approaches for ulnar nerve segmentation provides valuable insights for optimizing automated US-based nerve imaging. The findings suggest that traditional architectures with appropriate data augmentation can achieve robust performance. As these techniques mature, they have the potential to enhance clinical workflow efficiency, improve diagnostic accuracy, and ultimately benefit patient care in neuromuscular medicine.

## Figures and Tables

**Figure 1 medicina-62-00113-f001:**
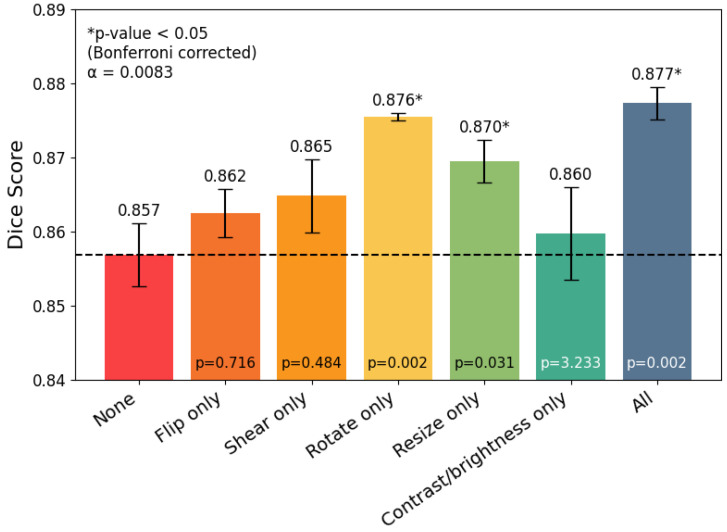
Comparison of the test Dice score when no data augmentations are applied, individual data augmentations are used, and all data augmentations are applied. Results are produced from 6 training runs for each scenario, totaling 42 trained U-Net models with a ResNet34 backbone.

**Figure 2 medicina-62-00113-f002:**
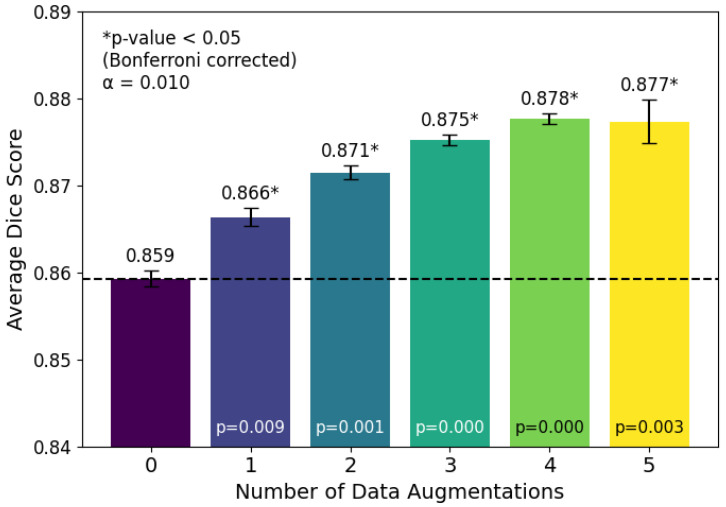
Average test Dice score by the number of data augmentations used. Results are averaged across three training sweeps for all 32 data augmentation combinations, yielding a total of 96 trained models, with the U-Net architecture using a ResNet34 backbone.

**Figure 3 medicina-62-00113-f003:**
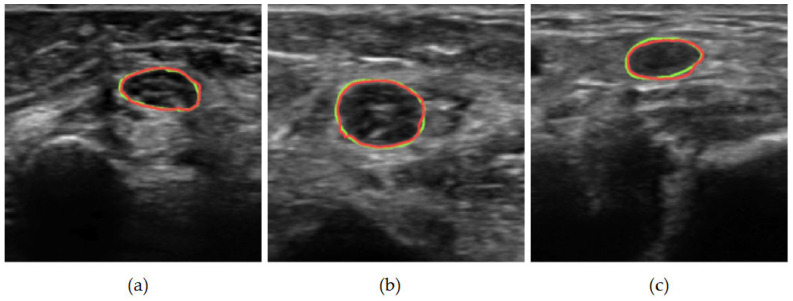
(**a**–**c**) General segmentation predictions of the ulnar nerve in US captured by miniSONO. Predictions are shown in red, and ground-truth labels are shown in green. Here the green segmentation represents the ground truth and the red segmentation represents the prediction made by the model.

**Figure 4 medicina-62-00113-f004:**
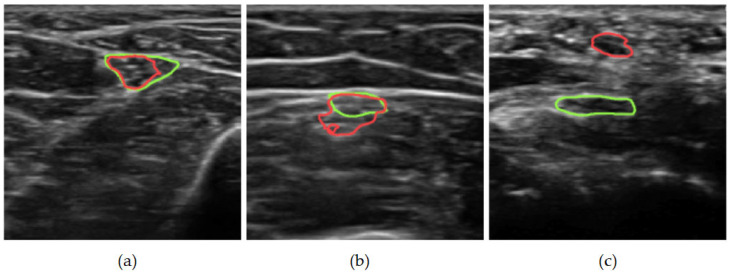
(**a**–**c**) Examples of segmentation predictions where the nerve was not identified well or at all. Here the green segmentation represents the ground truth and the red segmentation represents the prediction made by the model. Here the green segmentation represents the ground truth and the red segmentation represents the prediction made by the model.

**Table 1 medicina-62-00113-t001:** The computational complexity of each model tested in this study for an input size of 512 × 512.

Architecture	ResNet18	ResNet34	MiT-b0	MiT-b1	Efficientnet-b0	Efficientnet-b1
U-Net	21 G, 14 M	31 G, 24 M	—	—	10 G, 2 M	10 G, 2 M
U-Net++	64 G, 16 M	73 G, 26 M	—	—	20 G, 3 M	20 G, 3 M
DeepLabV3+	18 G, 12 M	31 G, 22 M	—	—	2 G, 939 K	2 G, 959 K
LinkNet	12 G, 12 M	21 G, 22 M	—	—	823 M, 199 K	878 M, 219 K
PAN	16 G, 11 M	29 G, 21 M	—	—	392 M, 131 K	467 M, 151 K
Segformer	—	—	7 G, 4 M	13 G, 14 M	—	—

MACs given in giga (G) and mega (M), and parameter size given in millions (M) and thousands (K).

**Table 2 medicina-62-00113-t002:** Comparison of segmentation performance across the different architectures considered in this study. Bold text indicates the best value for a given metric.

Architecture	Backbone	Recall	Precision	F1	Dice	IoU
U-Net	ResNet18	0.8447	0.897	0.8701	0.8679	0.7917
U-Net	ResNet34	0.869	0.8919	**0.8803**	**0.8808**	**0.8063**
U-Net	EfficientNet-b0	0.8609	0.8846	0.8725	0.8698	0.7951
U-Net	EfficientNet-b1	0.8466	0.9021	0.8735	0.8678	0.7898
U-Net++	ResNet18	0.846	0.9003	0.8723	0.8665	0.7925
U-Net++	ResNet34	0.8609	0.8912	**0.8758**	0.8746	0.8015
U-Net++	EfficientNet-b0	0.842	0.8948	0.8676	0.868	0.7913
U-Net++	EfficientNet-b1	0.8322	0.9002	0.8648	0.8559	0.7796
DeepLabV3+	ResNet18	0.8482	0.8946	0.8708	0.8659	0.7889
DeepLabV3+	ResNet34	0.8619	0.8882	0.8749	0.8737	0.7987
DeepLabV3+	EfficientNet-b0	0.8551	0.8773	0.866	0.863	0.7804
DeepLabV3+	EfficientNet-b1	0.8485	0.8864	0.867	0.8645	0.7834
Linknet	ResNet18	0.8517	0.8926	0.8717	0.8699	0.792
Linknet	ResNet34	0.8497	0.8853	0.8672	0.868	0.7913
Linknet	EfficientNet-b0	0.8437	0.8776	0.8603	0.8598	0.7772
Linknet	EfficientNet-b1	0.841	0.8735	0.8569	0.8581	0.7767
PAN	ResNet18	0.8163	**0.9112**	0.8612	0.8483	0.7658
PAN	ResNet34	0.8549	0.8909	0.8725	0.8727	0.7936
PAN	EfficientNet-b0	0.8057	0.9061	0.8529	0.8444	0.7545
PAN	EfficientNet-b1	0.8339	0.893	0.8625	0.8565	0.7722
Segformer	MIT-b0	0.8594	0.8515	0.8554	0.8578	0.7754
Segformer	MIT-b1	**0.8659**	0.8717	0.8688	0.8685	0.7906

## Data Availability

The datasets generated and/or analyzed during the current study are not publicly available due to a lack of IRB approval for public release, but are available from the corresponding author on reasonable request.

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
