# Peer review of "Deep Learning-Based Segmentation of the Ulnar Nerve in Ultrasound Images"

_medicina, 2026, doi:10.3390/medicina62010113_

Round 1

Reviewer 1 Report

Comments and Suggestions for Authors

Clarify whether the dataset includes both proximal (elbow) and distal (wrist) ulnar nerve regions, and if models performed equally across them.

The study only includes subjects with normal conduction studies, this should be explicitely stated in limitations and discussed how pathology (e.g., nerve swelling, altered echotexture in entrapment neuropathy) may alter model performance.

The discussion is somehow vague, particularly in term of clinical applications and future directions. I suggest authors to comment on how histological studies with cross-referencing the fascicles between modalities such as histology and US could aid in more accurate recognition of nerve fascicles on US, which is important in pathologies. See and reference article: https://www.nature.com/articles/s41598-024-84396-y

The vast of similar studies are made on median nerve as most frequently injured in CTS. Include a quantitative or at least contextual comparison to existing median nerve segmentation studies!

Author Response

Comments 1: Clarify whether the dataset includes both proximal (elbow) and distal (wrist) ulnar nerve regions, and if models performed equally across them.

Response 1: Thanks for your comment. I agree with your feedback that clarification is needed regarding the dataset description.

The dataset for this study included only the proximal (elbow) regions of the ulnar nerve. We collected ulnar nerve data only at the elbow because this is where most clinical problems occur. In cases of cubital tunnel syndrome, nerve compression happens near the medial epicondyle of the humerus. Therefore, we collected images at five points, both proximal and distal to the medial epicondyle, at 1-cm intervals. Consequently, we revised the methods section accordingly.

Comments 2: The discussion is somehow vague, particularly in term of clinical applications and future directions. I suggest authors to comment on how histological studies with cross-referencing the fascicles between modalities such as histology and US could aid in more accurate recognition of nerve fascicles on US, which is important in pathologies. See and reference article: https://www.nature.com/articles/s41598-024-84396-y

Response 2: Thank you very much for this insightful and constructive suggestion. We agree that a more precise explanation of the relationship between histologic validation and ultrasound-based nerve segmentation would strengthen the clinical relevance and future direction of our study.

In response, we have substantially expanded the Discussion to address how intrinsic limitations of peripheral nerve ultrasound—such as indistinct fascicular boundaries, difficulty differentiating perineurium and epineurium, and variability in intraneural echotexture—can hinder accurate segmentation. We explicitly highlight how these limitations were reflected in our own failure cases (e.g., Figure 4), in which the model misinterpreted perineurial tissue or adjacent hyperechoic structures as nerve margins.

Guided by the findings of Pušnik et al., we have also added a detailed explanation of how histology–ultrasound multimodal studies reveal that structures appearing as single fascicles on ultrasound may represent clusters of multiple fascicles histologically, and that fascicular boundaries can shift or blend based on scanning angle, probe pressure, or tissue composition.

Following your advice, we now discuss how incorporating histologic ground truth or MR microscopy–derived fascicular maps could improve model robustness, refine nerve boundary detection, and ultimately support fascicle-level interpretation—particularly in neuropathic conditions where fascicular changes are clinically meaningful.

Finally, we expanded the future-direction section to emphasize the importance of multimodal fusion approaches that integrate histologic insights into deep-learning frameworks. These revisions clarify how such approaches may overcome the intrinsic resolution limits of ultrasound, provide biologically grounded constraints for segmentation, and enhance the clinical applicability of automated nerve imaging.

All new content has been added to the Discussion section and highlighted in the revised manuscript.

Comments 3: The vast of similar studies are made on median nerve as most frequently injured in CTS. Include a quantitative or at least contextual comparison to existing median nerve segmentation studies!

Response 3: Thank you for this valuable comment. We agree that comparing our findings with prior work on median nerve segmentation helps contextualize the contribution of our study, particularly given that most deep-learning–based peripheral nerve ultrasound studies have focused on the median nerve.

To address your suggestion, we have revised the Discussion to include a contextual comparison with previously published median nerve segmentation research. Prior studies have generally reported Dice coefficients in the range of 0.80–0.90 depending on model architecture and dataset characteristics, as cited in references [18,19,35,36,40]. Our best-performing model achieved a Dice score of 0.8808 and an IoU of 0.8063, placing its performance within the upper range of these earlier median nerve studies. Notably, despite the ulnar nerve exhibiting greater anatomical variability and more heterogeneous echotexture than the median nerve, our model achieved a comparable level of accuracy. We now highlight that this finding suggests that, with sufficiently large datasets and appropriate augmentation strategies, ulnar nerve segmentation can attain accuracy similar to that of the median nerve.

This comparison clarifies the position of our work within the existing literature and underscores its contribution in addressing the relative scarcity of ulnar nerve segmentation research. The revised text has been added to the Discussion and is highlighted in the updated manuscript.

Reviewer 2 Report

Comments and Suggestions for Authors

This is an important study. Its strengths are: 

Largest dataset for ulnar nerve segmentation reported to date

Systematic evaluation of multiple architectures and augmentation strategies

Practical insights on model selection (simpler models sufficient)

Honest reporting of limitations and failure cases

Strong technical execution with appropriate methodology

Clinical relevance - addresses real clinical need

Impact:

Will serve as important benchmark for future ulnar nerve segmentation work

Provides valuable guidance on data augmentation for ultrasound imaging

Demonstrates that sophisticated architectures may not be necessary with adequate data

Establishes foundation for future clinical translation efforts.  Having said that, I believe it could be improved by some revisions: 

I Suggest this revisions:

Clarify experimental design (Data Augmentation):

Resolve inconsistencies between Figures 1 and 2

Clearly describe how many model training runs were performed

Explain the relationship between single augmentation tests and full combinatorial tests

Improve statistical reporting:

Report actual p-values

Apply multiple comparison correction

Add confidence intervals to all metrics

Clarify variance in Table 2 results

Expand limitations discussion:

Emphasize that only NORMAL nerves were studied

Discuss implications for clinical deployment

Add limitation of single annotator

Discuss missing CSA measurement validation

Strengthen clinical relevance discussion:

How does performance translate to clinical utility?

What accuracy is needed for clinical decision-making?

Compare to inter-observer variability if available

Discuss next steps for clinical validation

Methods clarification:

Describe annotation protocol in detail

Clarify validation strategy during training

Specify data quality control procedures

Author Response

Comments 1: Clarify experimental design (Data Augmentation):

Comments 1a: Resolve inconsistencies between Figures 1 and 2

Response 1a: We have updated both figures to for consistency and reporting of p-values. If the reviewers have any additional feedback on how may improve our figures, it would be greatly appreciated.

Comments 1b: Clearly describe how many model training runs were performed

Response 1b: The number of training runs for the augmentations are described in both figure 1 and 2.

Comments 1c: Explain the relationship between single augmentation tests and full combinatorial tests

Response 1c: To clarify this point we have added the following description to our results section:
  The results in Figure 1 show the impact of individual augmentations, with the most significant being the rotation augmentation, while Figure two shows that the number of combinations employed is also statistically significant in improving overall segmentation performance.

Comments 2: Improve statistical reporting:

Comments 2a: Apply multiple comparison correction

Response 2a: Thank you for bring up this important aspect of statistical analysis when doing multiple comparisons. Both Figure 1 and Figure 2 have been updated to include the raw p-values after applying Bonferroni correction. We add the following explanation to the Methods and the Results sections.

Methods:

For our augmentation analysis, we employ Bonferroni correction to our paired t-tests. Bonferroni correction is a statistical adjustment that accounts for the increased risk of false positives when performing multiple comparisons simultaneously.

Results:

For both Figure 1 and Figure 2, the overall pattern of statistical significance did not change after applying this correction.

Comments 3: Expand limitations discussion:

Comments 3a: Discuss implications for clinical deployment

Response 3a: Thank you for this important comment. We have expanded the Discussion to address the implications for clinical deployment of our model. Specifically, we added a new paragraph noting that real-world implementation will require validation across different ultrasound devices, transducer types, and operators, as image characteristics may vary substantially outside the controlled conditions of our study. We also emphasize the need to evaluate real-time inference feasibility and to assess performance in patients with neuropathies, whose nerve morphology often differs from that of healthy individuals. These revisions clarify that additional steps are necessary before the model can be integrated into clinical workflows. The new content has been added to the Discussion section and is highlighted in the revised manuscript.

Comments 3b: Add limitation of single annotator

Response 3b: Thank you for pointing this out. As noted in the original manuscript, all ultrasound examinations and annotations were performed by a single expert examiner. To better address this limitation, we have expanded the Discussion and Limitations sections to explicitly state that single-annotator labeling does not capture inter-observer variability, which is intrinsic to neuromuscular ultrasound. We also note that future work should include multiple annotators or consensus labeling to improve the robustness and generalizability of the ground truth. The revised text is now highlighted in the manuscript.

Comments 3c: Discuss missing CSA measurement validation

Response 3c: Thank you for this valuable comment. We agree that evaluating clinically relevant metrics, such as cross-sectional area (CSA), is essential for assessing the clinical utility of segmentation models. As noted in the revised manuscript, our study focused solely on segmentation performance and did not directly assess CSA measurement accuracy. We have clarified this limitation in both the Discussion and Limitations sections and emphasized that future work should evaluate how automated segmentation influences CSA accuracy and diagnostic decision-making in neuromuscular ultrasound. These revisions have been added to the manuscript and are highlighted accordingly.

Comments 4: Strengthen clinical relevance discussion:

Comments 4a: How does performance translate to clinical utility?

Response 4a: Thank you for this insightful comment. We have expanded the Discussion to clarify how the performance of our segmentation model relates to clinical utility. The Dice score of 0.8808 and IoU of 0.8063 achieved by our best-performing model are comparable to levels of inter-observer variability reported in neuromuscular ultrasound, suggesting that the model can delineate nerve boundaries with an accuracy similar to that of experienced clinicians. Given the anatomical variability and heterogeneous echotexture of the ulnar nerve, this level of performance indicates strong potential to support clinical workflows.

In clinical practice, reliable nerve boundary segmentation underpins automated cross-sectional area (CSA) measurement, the detection of focal nerve enlargement, and the quantitative monitoring of morphological changes over time. Because CSA is a primary diagnostic marker of ulnar neuropathy, consistent boundary identification by an automated model represents an important step toward clinically meaningful automation. However, we also note that segmentation accuracy does not necessarily guarantee accurate CSA estimation, and future validation studies are needed to determine how well automated segmentation supports diagnostic decision-making.

These points have been added to the revised Discussion and are highlighted in the updated manuscript.

Comments 4b: What accuracy is needed for clinical decision-making?

Response 4b: Thank you for this critical question. We have expanded the Discussion to clarify what level of accuracy is generally considered acceptable for clinical decision-making in neuromuscular ultrasound. Prior studies have shown that inter-observer variability in peripheral nerve ultrasound typically corresponds to Dice coefficients of approximately 0.80–0.88 when different clinicians manually delineate nerve boundaries. This indicates that even among experienced examiners, nerve contour definition shows some variability.

In this context, the Dice score of 0.8808 achieved by our best-performing model falls within — and at the upper end of — the range of accuracy associated with clinically acceptable human performance. Therefore, our model achieves segmentation accuracy sufficient to support downstream clinical tasks, such as automated CSA measurement and detection of focal nerve enlargement.

We also note, however, that segmentation accuracy alone does not guarantee clinical reliability, and future studies should explicitly evaluate the diagnostic performance of automated CSA measurements derived from the segmentation output.

These considerations have been incorporated into the Discussion and are highlighted in the revised manuscript.

Comments 4c: Compare to inter-observer variability if available

Response 4c: Thank you for this valuable comment. Because a single expert examiner generated all annotations in our dataset, our study did not include multiple annotators and therefore cannot directly quantify inter-observer variability. We have clarified this limitation in the Discussion and Limitations sections.

However, prior studies have reported that inter-observer variability in neuromuscular ultrasound typically corresponds to Dice coefficients of approximately 0.80–0.88 when different clinicians manually delineate nerve boundaries. Based on these published ranges, the Dice score of 0.8808 achieved by our best-performing model falls at the upper end of human-level variability. This suggests that, although we cannot compute inter-observer variability within our dataset, the segmentation performance of our model is comparable to clinically acceptable levels of human agreement. These points have been incorporated into the revised Discussion.

Comments 4d: Discuss next steps for clinical validation

Response 4d: Thank you for this critical comment. We have expanded the Discussion to more clearly outline the next steps required for clinical validation of our model. Specifically, we note that future studies should evaluate model performance across different ultrasound devices, transducer types, and operators, as image characteristics can vary considerably in real-world settings. Furthermore, because our dataset included only normal ulnar nerves, validation in patients with neuropathies or entrapment syndromes will be essential to determine whether the model can reliably detect or segment pathological nerves.

We also highlight the need to assess whether automated segmentation supports accurate calculation of clinically relevant metrics such as cross-sectional area (CSA), which is central to neuromuscular diagnostic workflows.

Finally, practical deployment will require evaluating real-time inference capability and integrating it into routine ultrasound workflows. These points have been added to the revised Discussion and have been highlighted in the updated manuscript.

Comments 5: Methods clarification:

Comments 5a: Describe annotation protocol in detail

Response 5a: Thank you for this helpful comment. We have expanded the Methods section to provide a more detailed description of the annotation protocol. In our study, nerve boundaries were defined using the inner epineurial contour, rather than the outer hyperechoic rim. This means that the segmentation mask followed the inner boundary of the epineurium to represent neural tissue better and excluded the outer hyperechoic connective tissue. Ambiguous regions—particularly where the inner epineurium blended with surrounding hyperechoic structures—were resolved by reviewing adjacent frames and applying consistent labeling rules. We also specify the annotator’s level of experience, the labeling software used, and the quality-control steps applied to all masks. These revisions have been added to the updated manuscript and are highlighted accordingly.

Comments 5b: Clarify validation strategy during training

Response 5b: Thank you for this helpful comment. We have revised the Methods section to describe the validation strategy used during model development clearly. To prevent data leakage, all dataset splits were performed patient-wise, ensuring that images from the same subject did not appear in the training, validation, and test sets. Data augmentations were applied only during training, and validation performance was monitored throughout training to guide learning-rate scheduling and assess convergence. These clarifications have been added to Section 2.5 (Implementation Details) and are highlighted in the updated manuscript.

Comments 5c: Specify data quality control procedures

Response 5c: Thank you for this comment. We have added explicit details regarding the data quality control procedures to the Methods section. Images exhibiting severe motion artifacts, acoustic shadowing, signal dropout, or excessive deformation due to probe pressure were excluded from the dataset. All remaining images underwent expert review prior to annotation to ensure adequate visualization of the ulnar nerve and overall suitability for segmentation.

These details have been incorporated into the Dataset Description section and are highlighted in the revised manuscript.

Round 2

Reviewer 1 Report

Comments and Suggestions for Authors

Authors improved the manuscript, well done